# Reflow Soldering Capability Improvement by Utilizing TaN Interfacial Layer in 1Mbit RRAM Chip

**DOI:** 10.3390/mi13040567

**Published:** 2022-03-31

**Authors:** Peng Yuan, Danian Dong, Xu Zheng, Guozhong Xing, Xiaoxin Xu

**Affiliations:** 1Key Laboratory of Microelectronics Device & Integrated Technology, Institute of Microelectronics, Chinese Academy of Sciences, Beijing 100190, China; yuanpeng@ime.ac.cn (P.Y.); dongdanian@ime.ac.cn (D.D.); zhengxu@ime.ac.cn (X.Z.); gzxing@ime.ac.cn (G.X.); 2School of Microelectronics, University of Chinese Academy of Sciences, Beijing 100029, China

**Keywords:** RRAM, thermal stability, reflow soldering, retention

## Abstract

We investigated the thermal stability of a 1Mbit OxRRAM array embedded in 28 nm COMS technology. A back-end-of-line (BEOL) solution with a TaN–Ta interfacial layer was proposed to eliminate the failure rate after reflow soldering assembly at 260 °C. By utilizing a TaN–Ta interfacial layer (IL), the oxygen defects in conductive filament were redistributed, and electromigration lifetimes of Cu-based damascene interconnects were improved, which contributed to optimization. This work provides a potential solution for the practical application of embedded RRAM beyond the 28 nm technology node.

## 1. Introduction

With the development of logic technology, embedded flash memory has become extremely complex and expensive when it is scaled below the 28 nm technology node. As a back end of line (BEOL) based emerging memory, resistive random access memory (RRAM) has been proven to be a possible alternative in encryption, code storage, and neuromorphic hardware systems, etc. due to its nonvolatile functionality, complementary metal-oxide semiconductor transistor (CMOS) process compatibility, and scalability with three-dimensional architecture [1,2,3]. For production-grade embedded applications, the program code and weight can be loaded in an embedded memory chip or in deep learning inference engines prior to reflow soldering [4]. The tail bits occurring in the reflow process can lead to greater reliability issues in long-term application. Therefore, the capabilities of reflow solder assembly, which involves connecting components to a printed circuit board (PCB) at the extreme temperature of 260 °C, must be considered. State-of-the-art reflow-capable STT-MRAM and FRAM have been reported [5,6]. However, the reflow soldering process for RRAM chips is lacking in presentation and research. 

We reported on a 1Mbit RRAM array fabricated on a 28 nm CMOS platform, which showed a barely satisfied reflow soldering tolerance [7]. The obviously degraded resistance during reflow caused the risk of retention failure, subsequently. Hence, we introduce a CMOS-applicable interfacial engineering of the TaN–Ta interfacial layer to improve thermal stability during the reflow process. This solution eliminates the failure rate of the 260 °C reflow test effectively. Moreover, the over-programmed low resistance state (LRS) degradation can be restrained in the TaN–Ta interfacial layer stack by adjusting the initially formed filament. The data retention characteristics after 260 °C reflow were investigated three times, proving a high thermal stability performance by this optimized TaN–Ta interfacial engineering. 

## 2. Experiment

Figure 1a shows the layout of a 1 Mbit embedded RRAM in 28 nm technology. Figure 1b displays a cross-sectional view of the RRAM array. Figure 1c shows the schematic diagram of the one-transistor-one-resistor (1T1R) cell structure. The RRAM cell was integrated on the drain electrode of the transistor. The process flow of the integration of RRAM was described in the previous work [7]. The RRAM cell was integrated between a tungsten plug and metal one (M1), and only one noncritical extra mask was needed. The TMO_x_-switching layer and top electrode were deposited in the M1 trench, sequentially. A Ta interfacial layer was deposited between the top electrode and TMO_x_-switching layer. A tungsten plug with a diameter of 80 nm served as the bottom electrode, which defined the area of the RRAM cell. The fabrication of the RRAM was independent with front-end-of-line (FEOL) CMOS technology, so it was capable of advanced technology, such as HKMG and FiFET. In the memory array, the drain electrodes of the 1T1R in the same row were connected to form bit lines (BLs). The word lines (WLs) were formed by connecting the gate electrodes in the same column. We used a common source configuration in the 2T2R structure to improve the storage density of the memory array. The program scheme, including forming, set, and reset, for the memory cell is shown in Figure 1c. Figure 1d shows the typical IV characteristics of the RRAM unit. During the program, the compliance current was modulated by the gate voltage of the transistor. The forming voltage was between 2 and 3 V, and the set and reset voltages were below 1.5 V, which was compatible with the 28 nm logic device. A custom-made automatic tester based on a Keithley 4200 semiconductor analyzer was utilized for array-level electrical testing.

## 3. Results and Discussion

Short exposure to extremely high temperature is common in microcontrollers with embedded non-volatile memory during solder assembly processes. Figure 2a shows the standard temperature profile during a reflow test, where the maximum temperature reached 260 °C, following the international JEDEC standard [8]. To illustrate the bit errors that may occur during the post-assembly process, we investigated the thermal stability of a 1Mbit RRAM chip qualified for reflow soldering using a custom-made RRAM array testing system. We divided a RRAM cell in sub-array (10 Kbit) into two groups and programmed them to high- and low-resistance states. All the RRAM cells were baked three times for a short bake duration (5 min) at 260 °C to simulate the solder reflow process. A read voltage of 0.3 V was applied on the BL to check the resistance of the cells. Figure 2b shows the high-resistance state (HRS) and LRS distribution before and after three reflow cycles. There were slight tail bits up to about 1% for the HRS after the reflow process, and the LRS tended to hardly decrease. These tail bits were the weak points for the subsequent application, posing a reliability challenge for the embedded memory.

To improve the reliability of the 1M RRAM array, we developed a simple BEOL process that added a thin TaN IL between the Ta layer and the top electrode, as shown in Figure 3a. Inserting the barrier layer proved to be an effective way to suppress the negative SET behavior and electrode ion diffusion [9,10]. By depositing the TaN interfacial layer, the generation of oxygen vacancies in the active layer was restrained due to the poor oxygen absorption of TaN compared with Ta. This explanation can be confirmed by Figure 3b. The LRS of devices with a TaN layer were two times larger than those without a TaN layer. In two cases, these cells were programmed by identical set schemes. Figure 3c shows the HRS and LRS distributions before and after reflow and the thermal budget for a RRAM array with the TaN IL. Compared to RRAM without the TaN IL, the tails bits of the HRS distribution were significantly reduced. The devices showed almost no degradation during solder reflow, providing a large margin for subsequent reliability immunity in subsequent applications. 

For RRAM, the phenomenon of resistance switching arises from the formation and rupture of conductive filament (CF). The morphology of the filament is related to oxygen vacancy concentration, electric field, and thermal field [11]. According to our previous simulation [7], more Joule heat generated during the SET process will lead to lower oxygen vacancy concentration in the filaments. Ta and TaN have different oxygen absorption capacities, leading to different oxygen vacancy concentrations in the film. Considering this, we speculated that the size of the conductive filaments in the cells with a TaN–Ta IL stack would be thinner compared to that in the Ta-IL-stack cells. The HRS retention failure is schematically illustrated in Figure 3d,e. The appearance of the transformation from HRS to LRS at high temperature can be explained by that the oxygen ions (O^2−^) diffused laterally and vertically from the gap into the ambiance. The TaN IL effectively restrained the excessive formation of oxygen vacancy, and thinner conductive filaments were formed. Thinner residual filaments would reduce the possibility of formation of percolation in the gap region. Additionally, the selective Ta–TaN barrier in the metal interconnection was conducive to achieving lower defectivity in the reflow process and improving electromigration lifetimes. Copper diffusion, which may lead to the degradation of RRAM cells in a HRS, was inhibited [12]. The Cu-based damascene interconnect with a selectively deposited TaN barrier was still the main scheme BEOL interconnected below the 10 nm technology node. Therefore, the proposed interfacial engineering promoted the embedded application of RRAM under the 10 nm node. 

Some critical data written into chips before packaging, such as data for trimming and repairing, have more stringent requirements on thermal stability. Since these data do not change, the thermal stability of low-resistance devices requires more rigorous study. Since set pulse has a great influence on LRS retention [13], we investigated the relationship between retention characteristics and the set process by setting the pulse widths of the set program to 100 ns, 500 ns, 1 μs, 10 μs, 100 μs, and 500 μs. Figure 4a shows the failure rate of LRS obtained under various set conditions after baking for 30 h at 150 °C for RRAM cells without a TaN IL. The failure criterion was set as 200 KΩ. Insets show the distribution of LRS before and after baking. The failure rate of LRS rapidly increased to about 40% comparatively when the width of the program pulse rose from 10 μs (left) to 50 μs (right). The retention failure of LRS was dominated by oxygen recombination into the filaments of surrounding areas [11,14]. The size of the CF was determined by the programming conditions. According to the simulation developed in our previous work [7], it was confirmed that looser filaments are formed in the long-set process due to excess joule heat. The loose filaments resulted in a wider switching expansion inside the filament, which suffered more from O^2−^ out-diffusion from the ambient environment into the filament. Because the thermal conductivity of the TaN/Ta bilayer is better than that of the Ta layer, the TaN/Ta interlayer is benefit to the formation of dense conductive filament. As shown in Figure 4b, the above over-programming-induced LRS degradation disappeared by adding the TaN interfacial layer. Clearly, it can be observed that, even when the pulse width reached 500 μs, the obtained low resistance was still stable after baking at 150 °C for 30 h. Hence, the interfacial engineering of TaN–Ta layer optimized LRS retention significantly. In Figure 4c, we demonstrate the variation of the failure rate with respect to set pulse width in a more straightforward way. The insertion of a TaN layer eliminated LRS failure caused by over-programming. Moreover, a subsequent thermal stress test was performed. As shown in Figure 4d, the LRS of the device with a TaN IL showed only a few failed bits after 277 h of baking at a high temperature, exhibiting excellent thermal stability. These results indicate that the retention characteristics were improved and endured the practical application well for 28 nm RRAM.

## 4. Conclusions

In this paper, the thermal stability of pre-programmed RRAM was investigated systematically based on a 1T1R memory array fabricated on a 28 nm platform. To improve the capability of enduring reflow, a TaN interfacial layer was introduced. The adoption of TaN interfacial layer not only redistributed oxygen vacancies in the conductive filament, but also improved the electromigration lifetimes of Cu-based damascene interconnects, which was beneficial for improving the thermal stability of RRAM. Additionally, we further confirmed that the thermal stability of LRS was more immune to programming parameters when using this interfacial engineering. This optimized approach provides a possible solution for actual RRAM application.

## Figures and Tables

**Figure 1 micromachines-13-00567-f001:**
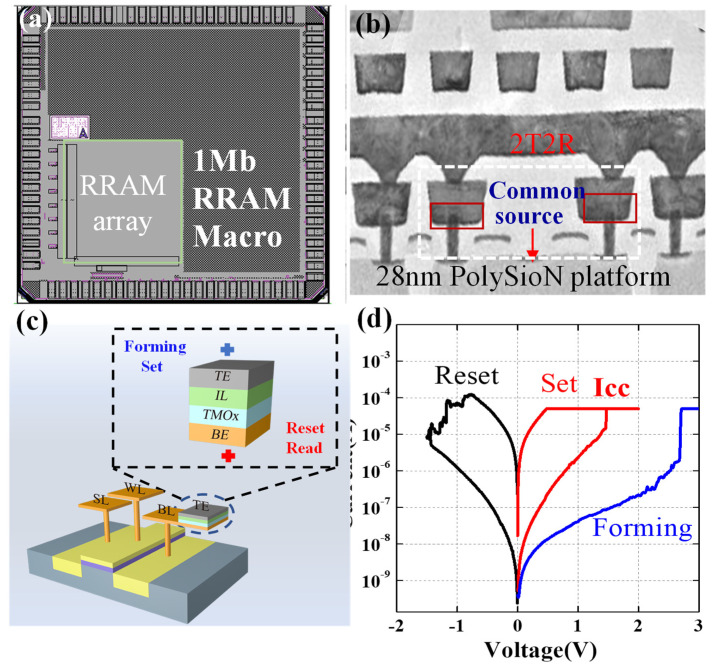
(**a**) Layout of 1Mb embedded RRAM macro. (**b**) Cross-sectional TEM image of RRAM cell. (**c**) Schematic diagram of the operation scheme of a TMOx-based RRAM device in a 1T1R configuration. (**d**) Typical IV curves of oxygen vacancy RRAM.

**Figure 2 micromachines-13-00567-f002:**
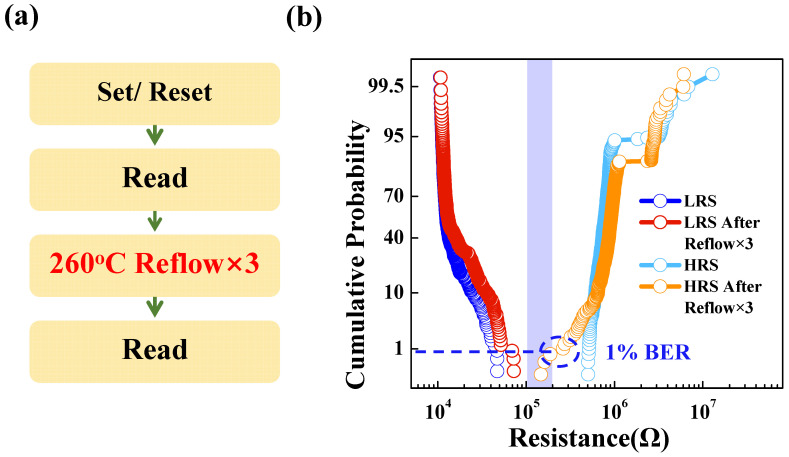
(**a**) The test flow chart of reflow. (**b**) The HRS and LRS distributions before and after the reflow process 3 times at 260 °C for Ta IL devices.

**Figure 3 micromachines-13-00567-f003:**
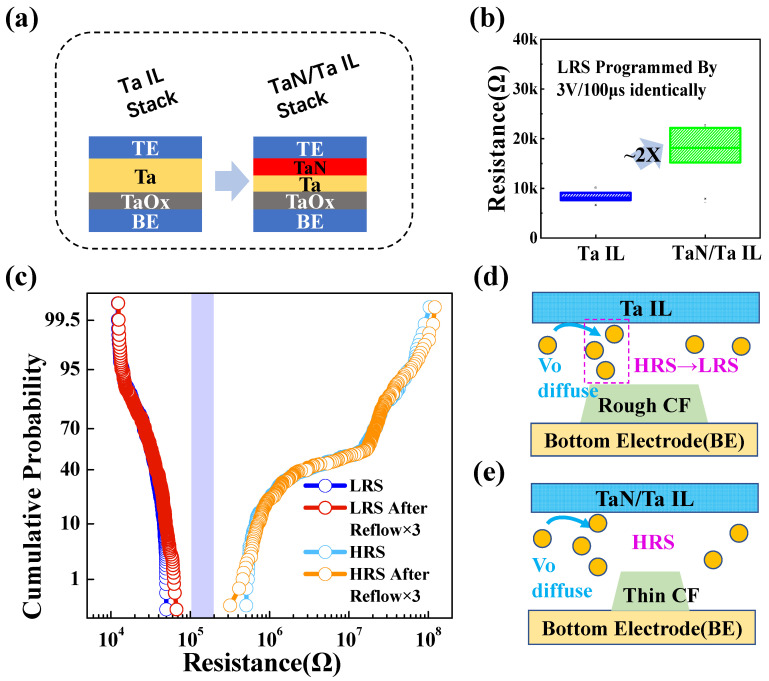
(**a**) The schematic diagram of the RRAM stack with an additional thin TaN layer to optimize retention behavior. (**b**) A comparison of LRS for the two structures described in (**a**). (**c**) The HRS and LRS distributions before and after the reflow process 3 times at 260 °C for TaN–Ta IL devices. The HRS retention failure scenarios of the structures with (**d**) the Ta IL and (**e**) the TaN–Ta IL.

**Figure 4 micromachines-13-00567-f004:**
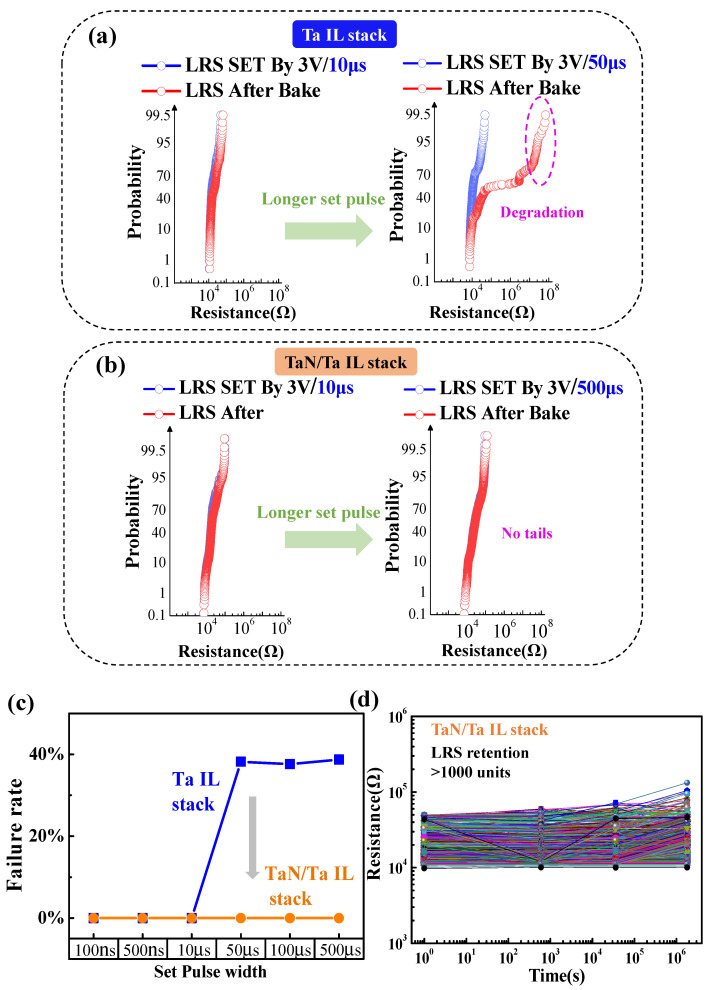
The LRS retention obtained under both a short set pulse and a long set pulse for (**a**) the Ta-IL-stack RRAM and (**b**) the TaN-Ta-IL-stack RRAM. (**c**) The trends of LRS failure rate variations at different set pulse widths. (**d**) LRS retention measurement of the TaN-Ta-IL-stack at 150 °C for 277 h with the forming pulse of 500 μs.

## Data Availability

The data that support the findings of this study are available from the corresponding author upon request.

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
