# Peer review of "Reflow Soldering Capability Improvement by Utilizing TaN Interfacial Layer in 1Mbit RRAM Chip"

_micromachines, 2022, doi:10.3390/mi13040567_

Round 1

Reviewer 1 Report

The work is well demonstrated.

The authors have presented the impact of TaN interfacial layer to improve the RRAM performance. I would suggest authors to have this clearly stated in the title and also include a comparison with state-of-art. 

Author Response

Reply to Comment 1: Thank you very much for your valuable comment.

According to your suggestion, we have changed our title to: Reflow Soldering Capability Improvement by Utilizing TaN Interfacial Layer in 1Mbit RRAM Chip

Besides, we investigated previous work and made a comparison with state-of-art.

  Approaches to address the thermal stability issue
Reference Device Method Merit Data retention improvement Reflow tolerance
[1] TiN/TaOx/HfO2/TiN Processing technology: annealing in the Oxygen atmosphere Simple but uncontrollable process Yes No comment
[2] TiN/TaO/AlOx/TiN Interfacial engineering: introduction of AlOx barrier layer Nonstandard CMOS materials Yes No comment
[3] Ir/Ta2O5-δ/TaOx/TaN Operation algorithm optimization: 2-step-forming Increased circuit consumption Yes No comment
[4] TiN\Ta2O5\Ta Processing technology: Annealing in NH3 atmosphere Non-standard annealing atmosphere Yes No comment
[5] HfO2 based RRAM Operation algorithm optimization: Pulse-width voltage-current write-verify-write (PVC-WVW) Increased circuit consumption Yes No comment
[6] HfO2 based RRAM Operation algorithm optimization: Low-voltage write-current-limiting-scheme Increased circuit consumption Yes No comment
This work W/TaN/Ta/TaOx/TiN Interfacial engineering: introduction of TaN/Ta barrier layer Standard CMOS materials Yes Yes

 [1] X. Huang, H. Wu, D. C. Sekar, S. N. Nguyen, K. Wang and H. Qian, "Optimization of TiN/TaOx/HfO2/TiN RRAM Arrays for Improved Switching and Data Retention," 2015 IEEE International Memory Workshop (IMW), 2015, pp. 1-4, doi: 10.1109/IMW.2015.7150300[2] Lin, YD., Chen, PS., Lee, HY. et al. Retention Model of TaO/HfOx and TaO/AlOx RRAM with Self-Rectifying Switch Characteristics. Nanoscale Res Lett 12, 407 (2017). doi: 10.1186/s11671-017-2179-5[3] T. Ninomiya, Z. Wei, S. Muraoka, R. Yasuhara, K. Katayama and T. Takagi, "Conductive Filament Scaling of TaOx Bipolar ReRAM for Improving Data Retention Under Low Operation Current," in IEEE Transactions on Electron Devices, vol. 60, no. 4, pp. 1384-1389, April 2013, doi: 10.1109/TED.2013.2248157[4] L. Goux et al., "Role of the Ta scavenger electrode in the excellent switching control and reliability of a scalable low-current operated TiN\Ta2O5\Ta RRAM device," 2014 Symposium on VLSI Technology (VLSI-Technology): Digest of Technical Papers, 2014, pp. 1-2, doi: 10.1109/VLSIT.2014.6894401.[5] P. Jain et al., "13.2 A 3.6Mb 10.1Mb/mm2 Embedded Non-Volatile ReRAM Macro in 22nm FinFET Technology with Adaptive Forming/Set/Reset Schemes Yielding Down to 0.5V with Sensing Time of 5ns at 0.7V," 2019 IEEE International Solid- State Circuits Conference - (ISSCC), 2019, pp. 212-214, doi: 10.1109/ISSCC.2019.8662393.[6] C. -C. Chou et al., "An N40 256K×44 embedded RRAM macro with SL-precharge SA and low-voltage current limiter to improve read and write performance," 2018 IEEE International Solid - State Circuits Conference - (ISSCC), 2018, pp. 478-480, doi: 10.1109/ISSCC.2018.8310392.

Reviewer 2 Report

The manuscript entitled “ Thermal Stability Improvement of 1Mbit RRAM Chip Qualified for Reflow Soldering at 260°C “ reports the enhanced thermal stability of the Ta/TaN interfacial layer for the protection of the pre-programmed state in comparison with Ta interfacial layer. The cumulative resistance state plot clearly shows the improved reliability for Ta/TaN interfacial layer. Due to the clear presentation of the data, there would be no doubt about the arguments. Short comments for the manuscript are given below : 

  1. The abbreviations ‘IL’ and ‘CF’ are not defined anywhere. Even though it may refer to ‘Interfacial layer’ and ‘conductive filament’, the abbreviation should be defined in the main text in their first usage. 
  2. The abbreviation ‘Vo’ is also not defined. The reviewer cannot recognize what it means. Do you mean ‘Oxygen Vacancy’? It should be defined in the main text at its first use stage. 
  3. The origin of performance enhancement is discussed in terms of the variation of oxygen vacancy related to the RESET state failure and Cu-based damascene interconnect. However, both arguments are only speculation-based opinions, and there is no clear experimental demonstration such as the cross-sectional elemental analysis from transmission (or scanning) electron microscope. Those studies are required for the justification of the author’s arguments for the origin of this phenomenon. 
  4. More controlled experimental conditions for the thermal inputs or the reflows might need to be investigated. In this manuscript, such thermal inputs are only given by the description of the general reflow process, which might give clear evidence for the physical origin of the Ta/TaN interfacial layer’s role.

In summary, either the microscopic analysis (such as element mapping) or the controlled macroscopic analysis (such as various thermal input conditions) might be required for a clear understanding of the phenomena related to the replacement of the Ta-only layer to Ta/TaN layer. 
